# Neural Mesh Flow: 3D Manifold Mesh Generation via Diffeomorphic Flows

**Kunal Gupta   Manmohan Chandraker**
University of California, San Diego
{k5gupta, mkchandraker}@eng.ucsd.edu

## Abstract

Meshes are important representations of physical 3D entities in the virtual world. Applications like rendering, simulations and 3D printing require meshes to be *manifold* so that they can interact with the world like the real objects they represent. Prior methods generate meshes with great geometric accuracy but poor *manifoldness*. In this work we propose Neural Mesh Flow (NMF) to generate two-manifold meshes for genus-0 shapes. Specifically, NMF is a shape auto-encoder consisting of several Neural Ordinary Differential Equation (NODE)[1] blocks that learn accurate mesh geometry by progressively deforming a spherical mesh. Training NMF is simpler compared to state-of-the-art methods since it does not require any explicit mesh-based regularization. Our experiments demonstrate that NMF facilitates several applications such as single-view mesh reconstruction, global shape parameterization, texture mapping, shape deformation and correspondence. Importantly, we demonstrate that manifold meshes generated using NMF are better-suited for physically-based rendering and simulation. Code and data are released.[1]

## 1  Introduction

Polygon meshes allow an efficient virtual representation of 3D objects, enabling applications in graphics rendering, simulations, modeling and manufacturing. Consequently, mesh generation or reconstruction from images or point sets has received significant recent attention. While prior approaches have primarily focused on obtaining geometrically accurate reconstructions, we posit that physically-based applications require meshes to also satisfy *manifold* properties. Intuitively, a mesh is manifold if it can be physically realized, for example, by 3D printing. Typically, reconstructed meshes are post-processed with humans in the loop for manifoldness, in order to enable ray tracing, slicing or Boolean operations. In contrast, we propose a novel deep network that directly generates manifold meshes (Fig. 1), alleviating the need for manual post-processing.

A manifold is a topological space that locally resembles Euclidean space in the neighbourhood of each point. A manifold mesh is a discretization of the manifold using a disjoint set of simple 2D polygons, such as triangles, which allows designing simulations, rendering and other manifold calculations. While a mesh data structure can simply be defined as a set $(\mathcal{V}, \mathcal{E}, \mathcal{F})$ of vertices $\mathcal{V}$ and corresponding edges $\mathcal{E}$ or face $\mathcal{F}$, not every mesh $(\mathcal{V}, \mathcal{E}, \mathcal{F})$ is manifold. Mathematically, we list various constraints on a singly connected mesh with the set $(\mathcal{V}, \mathcal{E}, \mathcal{F})$ that enables *manifoldness*[2].

- Each edge $e \in \mathcal{E}$ is common to exactly 2 faces in $\mathcal{F}$ (Fig. 2a)
- Each vertex $v \in \mathcal{V}$ is shared by exactly one group of connected faces (Fig. 2b)
- Adjacent faces $F_i, F_j$ have normals oriented in same direction (Fig. 2c)

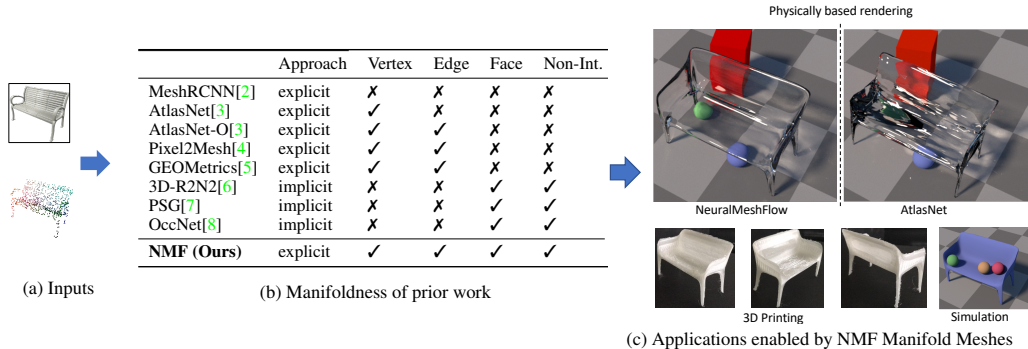

(a) Inputs

| | Approach | Vertex | Edge | Face | Non-Int. |
|---|---|---|---|---|---|
| MeshRCNN[2] | explicit | ✗ | ✗ | ✗ | ✗ |
| AtlasNet[3] | explicit | ✓ | ✗ | ✗ | ✗ |
| AtlasNet-O[3] | explicit | ✓ | ✓ | ✗ | ✗ |
| Pixel2Mesh[4] | explicit | ✓ | ✓ | ✗ | ✗ |
| GEOMetrics[5] | explicit | ✓ | ✓ | ✗ | ✗ |
| 3D-R2N2[6] | implicit | ✗ | ✗ | ✓ | ✓ |
| PSG[7] | implicit | ✗ | ✗ | ✓ | ✓ |
| OccNet[8] | implicit | ✗ | ✗ | ✓ | ✓ |
| **NMF (Ours)** | explicit | ✓ | ✓ | ✓ | ✓ |

(b) Manifoldness of prior work

(c) Applications enabled by NMF Manifold Meshes

Figure 1: Given an input as either a 2D image or a 3D point cloud (a) Existing methods generate corresponding 3D mesh that fail one or more manifoldness conditions (b) yielding unsatisfactory results for various applications including physically based rendering (c). NeuralMeshFlow generates manifold meshes which can directly be used for high resolution rendering, physics simulations (see supplementary video) and be 3D printed without the need for any prepossessing or repair effort.

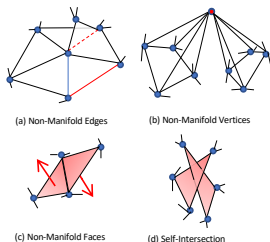

Figure 2: Non-manifold geometries for a part of singly connected mesh: (a) An edge that is shared by either exactly one (red) or more than two (red dashed) faces. (b) A vertex (red) shared by more than one group of connected faces. (c) Adjacent faces that have normals (red-arrow) oriented in opposite directions. (d) Faces intersecting other triangles of the same mesh.

The above mentioned constraints on a mesh $(\mathcal{V}, \mathcal{E}, \mathcal{F})$ guarantee it to be a manifold in the limit of infinitesimally small discretization. That is not the case when dealing with practical meshes with large and non-uniformly distributed triangles. To ensure physical realizability, we tighten the definition with a fourth practical constraint that no two triangles may *intersect* (Fig. 2d).

In this work, we pose the task of 3D shape generation as learning a diffeomorphic flow from a template genus-0 manifold mesh to a target mesh. Our key insight is that manifoldness is conserved under a diffeomorphic flow due to their uniqueness [9, 10] and orientation preserving property [11, 12]. In contrast to methods that learn "deformations" of a template manifold using an MLP or graph-based network [3–5], our approach ensures manifoldness of the generated mesh. We use Neural ODEs [1] to model the diffeomorphic flow, however, must overcome their limited capability to represent a wide variety of shapes [9, 10, 13], which has restricted prior works to single-category representations [14, 15]. We propose novel architectural features such as an instance normalization layer that enables generating 3D shapes across multiple categories and a series of diffeomorphic flows to gradually refine the generated mesh. We show quantitative comparisons to prior works and more importantly, compare resulting meshes on physically meaningful tasks such as rendering, simulation and 3D printing to highlight the importance of manifoldness.

**Toy example: regularizer's dilemma** Consider the task of deforming a template unit spherical mesh $S$ (Fig. 3a) into a target star mesh $T$ (Fig. 3b). We approximate the deformation with a multi-layer perceptron (MLP) $f_\theta$ with a unit hidden layer of 256 neurons with $relu$ and output layer with $tanh$ activation. We train $f_\theta$ by minimizing various losses over the points sampled from $S, T$. A conventional approach involves minimizing the Chamfer Distance $L_c$ between $S, T$, leading to accurate point predictions but several edge-intersections (Fig. 3c). By introducing edge length regularization [4] $L_e$, we get fewer edge-intersections (Fig. 3d) but the solution is also geometrically sub-optimal. We can further reduce edge-intersections with Laplacian regularization [4] (Fig. 3e), but this takes a bigger toll on geometric accuracy. Thus, attempting to reduce self-intersections by explicit regularization not only makes the optimization hard, but can also lead to predictions with lower geometric accuracy. In contrast, our proposed use of NODE (with dynamics $f_\theta$) is designed by construction [9, 10] to prevent self-intersections without explicit regularization (Fig. 3f).

In summary, we make the following contributions:

- A novel approach to 3D mesh generation, Neural Mesh Flow (NMF), with a series of NODEs that learn to deform a template mesh (ellipsoid) into a target mesh with greater *manifoldness*.

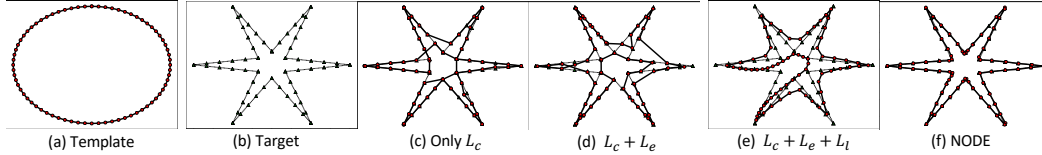

Figure 3: 2D Toy Example: For the task of deforming a manifold template mesh (a) to a target mesh (b) using explicit mesh regularization (c-e) trades edge-intersections for geometric accuracy . In contrast, a NODE [1] (f) is implicitly regularized preventing edge-intersections without loosing geometric accuracy.

- Extensive comparisons to state-of-the-art mesh generation methods for physically based rendering and simulation (see supplementary video), highlighting the advantage of NMF's manifoldness.
- New metrics to evaluate manifoldness of 3D meshes and demonstration of applications to single-view reconstruction, 3D deformation, global parameterization and correspondence.

## 2   Related Work

Existing learning based mesh generation methods, while yielding impressive geometric accuracy, do not satisfy one or more *manifoldness* conditions (Fig. 1b). While indirect approaches [6–8, 16–18] suffer from the non-manifoldness of the marching cube algorithm [19], direct methods [2–5] are faced with the *regularizer's dilemma* on the trade-off between geometric accuracy and higher manifoldness, illustrated in Fig. 3 and discussed in Sec. 1.

**Indirect Mesh Prediction**   Indirect approaches predict the 3D geometry as either a distribution of voxels [20–27], point clouds [7, 28] or an implicit function representing signed distance from the surface [8, 16, 18]. Both voxel and point set prediction methods struggle to generate high resolution outputs which later makes the iso-surface extraction tools ineffective or noisy [3]. Implicit methods feed a neural network with a latent code and a query point, encoding the spatial coordinates [8, 16, 18] or local features [29], to predict the TSDF value [16] or the binary occupancy of the point [8, 18]. However, these approaches are computationally expensive since in order to get a surface from the implicit function representation, several thousands of points must be sampled. Moreover, for shapes such as chairs that have thin structures, implicit methods often fail to produce a single connected component.

All the above methods depend on the marching cube algorithm [19] for iso-surface extraction. While marching cubes can be applied directly to voxel grids, point clouds first regress the iso-surface using surface normals. Implicit function representations must regress TSDF values per voxel and then perform extensive query to generate iso-surface based on a threshold $\tau$. This is used to classify grid vertices $v_i \in \mathcal{V}$ as 'inside' ($TSDF(v_i) \leq \tau$) and 'outside' ($TSDF(v_i) \geq \tau$). For each voxel, based on the arrangement of its grid vertices, marching cubes [19, 30–32] follows a lookup-table to find a triangle arrangement. Since this rasterization of iso-surface is a purely local operation, it often leads to ambiguities [30–32], resulting in meshes being *non-manifold*.

**Direct Mesh Prediction**   A mesh based representation stores the surface information cheaply as list of vertices and faces that respectively define the geometric and topological information. Early methods of mesh generation relied on predicting the parameters of category based mesh models [33–35]. While these methods output manifold meshes, they work only for object category with available parameterized manifold meshes. Recently, meshes have been successfully generated for a wide class of categories using topological priors [3, 4]. Deep networks are used to update the vertices of initial mesh to match that of the final mesh. AtlasNet [3] uses Chamfer distance applied on the vertices for training, while Pixel2Mesh [4] uses a coarse-to-fine deformation approach using vertex Chamfer loss. However, using a point set training scheme for meshes leads to severe topological issues and produced meshes are not manifold. Some recent works have proposed to use mesh regularizers like Laplacian [2, 4, 5, 36], edge lengths [2, 5], normal consistency [2] or pose it as a linear programming problem [37] to constrain the flexibilty of vertex predictions. They are either limited to low resolution meshes [37] or suffer from the *regularizer's dilemma* discussed in Fig. 3, as better geometric accuracy comes at a cost of manifoldness.

In contrast to the above approaches, the proposed NMF achieves high resolution meshes with a high degree of manifoldness across a wide variety of shape categories. Similar to previous approaches [3–5], an initial ellipsoid is deformed by updating its vertices. However, instead of using explicit mesh regularizers, NMF uses NODE blocks to learn the diffeomorphic flow to implicitly discourage

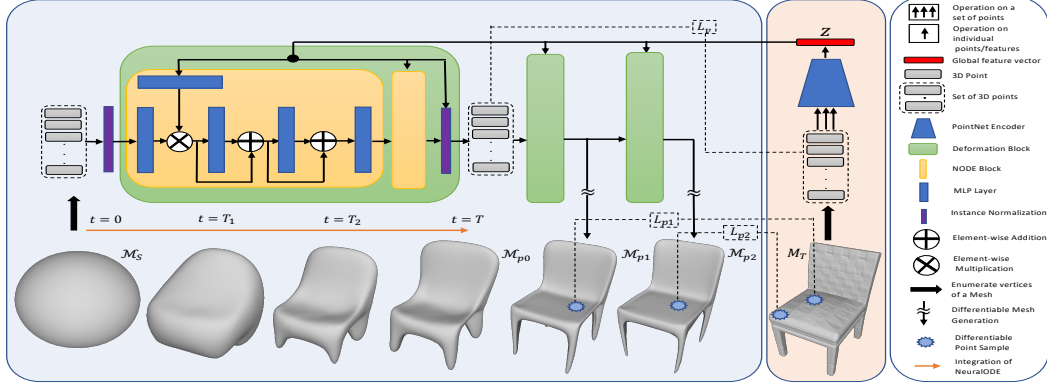

Figure 4: Neural Mesh Flow consists of three deformation blocks that perform point-wise flow on spherical mesh vertices based on the shape embedding $z$ from target shape $\mathcal{M}_T$. The bottom row shows an actual chair being generated at various stages of NMF. Time instances $0 < T_1 < T_2 < T$ show the deformation of spherical mesh into a coarse chair representation $\mathcal{M}_{p0}$ by the first deformation block. Further deformation blocks perform refinements to yield refined meshes $\mathcal{M}_{p1}, \mathcal{M}_{p2}$.

self-intersections, maintain the topology and thereby achieve better manifoldness of generated shape. The method is end-to-end trainable without requiring any post-processing.

## 3  Neural Mesh Flow

We now introduce Neural Mesh Flow (Fig 4), which learns to auto-encode 3D shapes. NMF broadly consists of four components. First, the target shape $\mathcal{M}_T$ is encoded by uniformly sampling $N$ points from its surface and feeding them to a PointNet [38] encoder to get the global shape embedding $z$ of size $k$. Second, NODE blocks diffeomorphically *flow* the vertices of template sphere towards target shape conditioned on shape embedding $z$. Third, the instance normalization layer performs non-uniform scaling of NODE output to ease cross-category training. Finally, refinement flows provide gradual improvement in quality. We start with a discussion of NODE and its regularizing property followed by details on each component.

**NODE Overview.**  A NODE learns a transformation $\phi_T : \mathcal{X} \to \mathcal{X}$ as solutions for initial value problem (IVP) of a parameterized ODE $x_T = \phi_T(x_0) = x_0 + \int_0^T f_\Theta(x_t)dt$. Here $x_0, x_T \in \mathcal{X} \subset \mathbb{R}^n$ respectively represent the input and output from the network with parameters $\Theta$ ($n = 3$ for our case), while $T \in \mathbb{R}$ is a hyper parameter that represents the duration of the *flow* from $x_0$ to $x_T$. For a well behaved dynamics $f_\Theta : \mathbb{R}^n \to \mathbb{R}^n$ that is Lipschitz continuous, any two distinct trajectories in $\mathbb{R}^n$ of NODE with duration $T$ may not intersect due to the existence and uniqueness of IVP solutions [9, 10]. Moreover, NODE manifests the orientation preserving property of diffeomorphic flows [11, 12]. These lead to strong implicit regularizations against self-intersection and non-manifold faces. There are several other advantages to NODE compared to traditional MLPs such as improved robustness [39], parameter efficiency [1], ability to learn normalizing flows [14, 15, 40] and homeomorphism [10]. We refer the readers to [9, 10, 13] for more details.

**Diffeomorphic Conditional Flow.**  The standard NODE [1] formulation cannot be used directly for the task of 3D mesh generation since they lack any means to feed in shape embedding and are therefore restricted to learning a few shape. A naive way would be to concatenate features to point coordinates like is done with traditional MLPs [4, 5] but this destroys the shape regularization properties due to several augmented dimensions [9, 10]. Our key insight is that instead of a fixed NODE dynamics $f_\Theta$ we can use a family of dynamics $f_{\Theta|z}$ parameterized by $z$ while still retaining the uniqueness property as long as $z$ is held constant for the purpose of solving IVP with initial conditions $\{x_0, x_T\}$.

The objective of conditional flow (NODE Block) therefore is to learn a mapping $F_{\Theta|z}$ (1) given the shape embedding $z$ and initial values $\{(p_I^i, p_O^i) | p_I^i \in \mathcal{M}_I, p_O^i \in \mathcal{M}_O\}$ where $\mathcal{M}_I, \mathcal{M}_O$ are respectively the input and output point clouds.

$$p_O^i = F_{\Theta|z}(p_I^i, z) = p_I^i + \int_0^T f_{\Theta|z}(p_I^i, z)dt \tag{1}$$

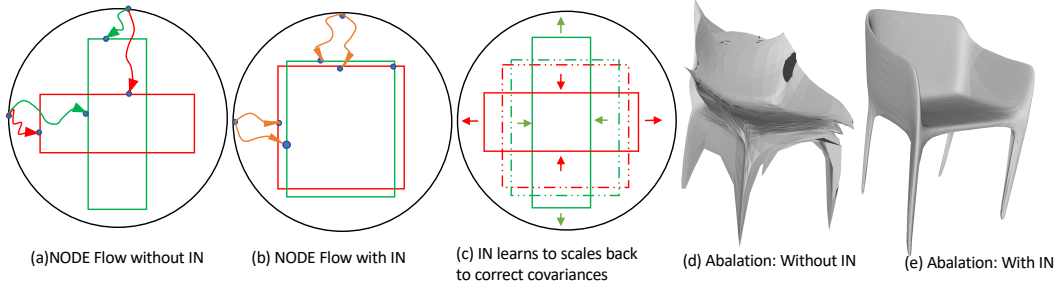

| (a)NODE Flow without IN | (b) NODE Flow with IN | (c) IN learns to scales back to correct covariances | (d) Abalation: Without IN | (e) Abalation: With IN |

Figure 5: The impact of instance normalization (IN) and refinement flows in NMF. (a) Learning deformation of a template (black) to target shapes of different variances (red and green) require longer non-uniform NODE trajectories making learning difficult. (b) IN allows NODE to learn deformations to an arbitrary variance. (c) This leads to simpler dynamics and can later be scaled back to correct shape variance. (d) A model trained without IN leads to self-intersections and non-manifold faces due to very complex dynamics being learned. (e) A model with IN is smoother and regularized.

|  | Chamfer-L2 ($\downarrow$) | Normal Consistency ($\uparrow$) | NM-Faces ($\downarrow$) | Self-Intersection ($\downarrow$) | Time ($\downarrow$) |
|---|---|---|---|---|---|
| No Instance Norm | 6.48 | 0.820 | 2.94 | 3.28 | 183 |
| 0 refinement | 5.00 | 0.818 | 0.39 | 0.03 | **68** |
| 1 refinement | 4.93 | 0.819 | **0.38** | **0.03** | 124 |
| 2 refinement | **4.65** | **0.818** | 0.73 | 0.09 | 189 |

Table 1: Ablation for Instance Normalization and refinement

**Instance Normalization.** Normalizing input and hidden features to zero mean and unit variance is important to reduce co-variate shift in deep networks [41–46]. While trying to deform a template sphere to targets with different variances (like a firearm and chair) different parts of the template need to be *flown* by very different amounts to different locations (Fig. 5a). This is observed to causes significant *strain* on the NODE which ends up learning more complex dynamics resulting in meshes with poor geometric accuracy and manifoldness (Fig 5d and Table 1). Instance normalization separates the task of learning target variances from that of learning target attributes. It gives NODE flexibility to deform the template to a target with arbitrary variance which yields better geometric accuracy(Fig. 5b). This is later scaled back to the correct variance by instance normalization layer (Fig. 5c) Given an input point cloud $\mathcal{M} \in \mathbb{R}^{N \times n}$ and its shape embedding $z$, the instance normalization calculates the point average $\mu \leftarrow \frac{1}{|\mathcal{M}|} \sum_i p^i, p^i \in \mathcal{M}$ and then applies non-uniform scaling $\mathcal{M} \leftarrow (\mathcal{M} - \mu) \odot \Delta(z)$ to arrive at correct target variances. Here $\Delta : \mathbb{R}^k \to \mathbb{R}^n$ is an MLP that regress variance coefficients for the $n$ dimensions based on shape embedding $z$. $\odot$ refers to the element wise multiplication.

**Overall Architecture.** A single NODE block is often not sufficient to get desired quality of results. We therefore stack up two NODE blocks in a sequence followed by an instance normalization layer and call the collection a deformation block. While a single deformation block is capable of achieving reasonable results (as shown by $\mathcal{M}_{p0}$ in Fig. 4) we get further refinement in quality by having two additional deformation blocks. Notice how the $\mathcal{M}_{p1}$ has a better geometric accuracy than $\mathcal{M}_{p0}$ and $\mathcal{M}_{p2}$ is *sharper* compared to $\mathcal{M}_{p1}$ with additional refinement. We report the geometric accuracy, manifoldness and inference time for different amounts of refinement in Table 1. The reported quantities are averaged over the 11 Shapnet categories (this excludes watercraft and lamp where NMF struggles with thin structures). For details on per category ablation, please see the supplementary material. To summarize, the entire NMF pipeline can be seen as three successive diffeomorphic flows $\{F^0_{\Theta|z}, F^1_{\Theta|z}, F^2_{\Theta|z}\}$ of the initial spherical mesh to gradually approach the final shape.

**Loss Function.** In order to learn the parameters $\Theta$ it is important to use a loss which meaningfully represents the difference between the predicted $M_P$ and the target $M_T$ meshes. To this end we use the bidirectional Chamfer Distance (2) on the points sampled differentiably [47] from predicted $\tilde{M}_P$ and target $\tilde{M}_T$ meshes.

$$\mathcal{L}(\Theta) = \sum_{p \in \tilde{M}_P} \min_{q \in \tilde{M}_T} ||p - q||^2 + \sum_{q \in \tilde{M}_T} \min_{p \in \tilde{M}_P} ||p - q||^2 \qquad (2)$$

We compute chamfer distances $\mathcal{L}_{p1}, \mathcal{L}_{p2}$ for meshes after deformation blocks $F^1_{\Theta|z}$ and $F^2_{\Theta|z}$. For meshes generated from $F^0_{\Theta|z}$ we found that computing chamfer distance $\mathcal{L}_v$ on the vertices gave

better results since it encourages predicted vertices to be more uniformly distributed (like points sampled from target mesh). We thus arrive at the overall loss function to train NMF.

$$\mathcal{L} = w_0\mathcal{L}_v + w_1\mathcal{L}_{p1} + w_2\mathcal{L}_{p2} \tag{3}$$

Here we take $w_0 = 0.1, w_1 = 0.2, w_3 = 0.7$ so as to enhance mesh prediction after each deformation block. The adjoint sensitivity [48] method is employed to perform the reverse-mode differentiation through the ODE solver and therefore learn the network parameters $\Theta$ using the standard gradient descent approaches.

**Dynamics Equation.** The Neural ODE $F_{\theta|z}$ is built around the dynamics equation $f_{\theta|z}$ that is learned by a deep network. Given a point $x \in \mathbb{R}^n$, we first get 512 length point features by applying a linear layer. To condition the NODE on shape embedding, we extract a 512 length shape feature from the shape embedding $z$ and multiply it element wise with the obtained point features to get the *point-shape* features. Thus, *point-shape* features contains both the point features as well as the global instance information. We find that dot multiplication of the shape and point features yields similar performance as their concatenation, albeit requiring less memory. Lastly, we feed the *point-shape* features into two residual MLP blocks each of width 512 and subsequent MLP of width 512 which outputs the predicted point location $y \in \mathbb{R}^n$. Based on the findings of [13, 49] we make use of the $tanh$ activation after adding the residual predictions at each step. This ensures maximum flexibility in the dynamics learned by the deep network. More details about the architecture can be found in supplementary material.

**Implementation Details** For the Neural Mesh Flow architecture, both the mesh vertices and NODE dynamics operate in $n = 3$ dimensions. We uniformly sample $N = 2520$ from the target mesh and using PointNet [38] encoder, get a shape embedding $z$ of size $k = 1000$. During training, the NODE is solved with a tolerance of $1e^{-5}$ and interval of integration set to $t = 0.2$ for deforming an icosphere with 622 vertices. The integration time was empirically determined to be large enough for flow to work, but not too large to cause overfitting. At test time, we use an icosphere of 2520 vertices and tolerance of $1e^{-5}$. We train NMF for 125 epochs using Adam [50] optimizer with a learning rate of $10^{-5}$, weight decay of $0.95$ after every 250 iterations and a batch size of 250, on 5 NVIDIA 2080Ti GPUs for 2 days. For single view reconstruction, we train an image to point cloud predictor network with pretrained ResNet [51] encoder of latent code 1000 and a fully-connected decoder with size 1000,1000,3072 with relu non-linearities. The point predictor is trained for 125 epochs on the same split as NMF auto-encoder.

## 4 Experiments

In this section we show qualitative and quantitative results on the task of auto-encoding and single view reconstruction of 3D shapes with comparison against several state of the art baselines. In addition to these tasks, we also demonstrate several additional features and applications of our approach including latent space interpolation texture mapping, consistent correspondence and shape deformations in the supplementary material.

**Data** We evaluate our approach on the ShapeNet Core dataset [52], which consists of 3D models across 13 object categories which are preprocessed with [53] to obtain manifold meshes. We use the training, validation and testing splits provided by [6] to be comparable to other baselines. We use rendered views from [6].

**Evaluation criteria** We evaluate the predicted shape $\mathcal{M}_P$ for geometric accuracy to the ground truth $\mathcal{M}_T$ as well as for manifoldness. For geometric accuracy, we follow [2] and compute the bidirectional Chamfer distance according to (2) and normal consistency using (4) on 10000 points sampled from each mesh. Since Chamfer distance is sensitive to the size of meshes, we scale the meshes to lie within a unit radius sphere. Chamfer distances are report by multiplying with $10^3$. With $\tilde{M}_P, \tilde{M}_T$ the point sets sampled from $\mathcal{M}_p, \mathcal{M}_T$ and $\Lambda_{P,Q} = \{(p, argmin_q||p - q||) : p \in P\}$, we define

$$\mathcal{L}_n = |\tilde{M}_P|^{-1} \sum_{(p,q)\in\Lambda_{\tilde{M}_P,\tilde{M}_T}} |u_p \cdot u_q| \ + \ |\tilde{M}_T|^{-1} \sum_{(p,q)\in\Lambda_{\tilde{M}_T,\tilde{M}_P}} |u_q \cdot u_p| \ - \ 1 \tag{4}$$

We detect non-manifold vertices (Fig. 2(b)) and edges (Fig. 2(a)) using [54] and report the metrics 'NM-vertices', 'NM-edges' respectively as the ratio($\times 10^5$) of number of non-manifold vertices and edges to total number of vertices and edges in a mesh. To calculate non-manifold faces, we count number of times adjacent face normals have a negative inner product, then the metric 'NM-Faces' is reported as its ratio(%) to the number of edges in the mesh. To calculate the number of instances

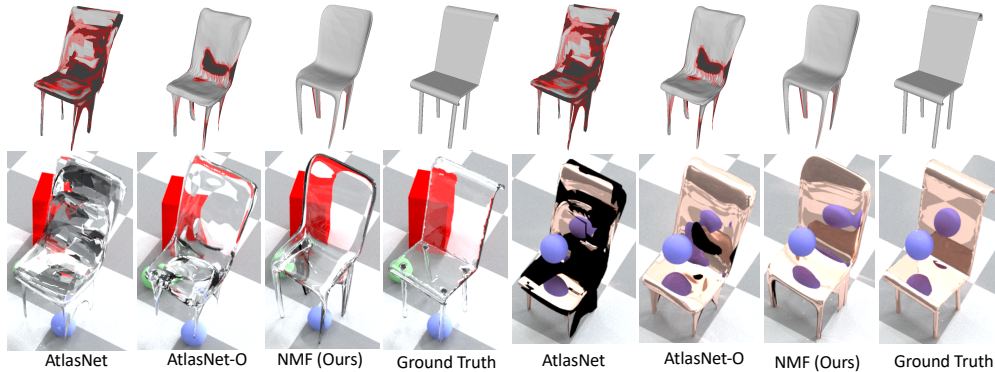

| | | | | AtlasNet | AtlasNet-O | NMF | Ground Truth |

AtlasNet AtlasNet-O NMF (Ours) Ground Truth            AtlasNet AtlasNet-O NMF (Ours) Ground Truth

Figure 6: Auto-encoder: The first row shows mesh geometry along with self-intersections (red) and flipped normals (black). The bottom row shows results from physically based rendering with dielectric and conducting materials. The appearances of the red box, green ball and blue ball are more realistic for NMF than AtlasNet, since the latter suffers from severe self-intersections and flipped normals.

| | Chamfer-L2 ($\downarrow$) | | | Normal Consistency ($\uparrow$) | | | NM-Faces ($\downarrow$) | | | Self-Intersection ($\downarrow$) | | |
|---|---|---|---|---|---|---|---|---|---|---|---|---|
| | AtNet | AtNet-O | NMF | AtNet | AtNet-O | NMF | AtNet | AtNet-O | NMF | AtNet | AtNet-O | NMF |
| mean | 4.15 | **3.50** | 5.54 | 0.815 | 0.816 | **0.826** | 1.72 | 1.43 | **0.71** | 24.80 | 6.03 | **0.10** |
| mean (with Laplace) | 4.59 | **3.81** | 5.25 | 0.807 | 0.811 | **0.826** | 0.47 | 0.56 | **0.38** | 13.26 | 2.02 | **0.00** |

Table 2: Auto-encoding performance.

of self-intersection, we use [55] and report the ratio(%) of number of intersecting triangles to total number of triangles in a mesh. Only the mean over all ShapeNet categories are reported in this paper with category specific details can be found in the supplementary.

For qualitative evaluation, we render predicted meshes via a physically based renderer [56] with dielectric and metallic materials to highlight artifacts due to non-manifoldness. While we render NMF meshes directly, other methods render poorly due to non-manifoldness and are smoothed prior to rendering to obtain better visualizations. Please see supplementary for further visualizations.

**Baselines** We compare with official implementations for Pixel2Mesh [2, 4], MeshRCNN [2] and AtlasNet [3]. We use pretrained models for all these baselines motioned in this paper since they share the same dataset split by [6]. We use the implementation of Pixel2Mesh provided by MeshRCNN, as it uses a deeper network that outperforms the original implementation. We also consider AtlasNet-O which is a baseline proposed in [3] that uses patches sampled from a spherical mesh, making it closer to our own choice of initial template mesh. We also create a baseline of our own called NMF-M, which is similar in architecture to NMF but trained with a larger icosphere of 2520 vertices, leading to slight differences in test time performance. To account for possible variation in manifoldness due to simple post processing techniques, we also report outputs of all mesh generation methods with 3 iterations of Laplacian smoothing. Further iterations of smoothing lead to loss of geometric accuracy without any substantial gain in manifoldness. We also compare with occupancy networks [8], a state-of-the-art indirect mesh generation method based on implicit surface representation. We compare with several variants of OccNet based on the resolution of Multi Iso-Surface Extraction algorithm [8]. To this end, we create OccNet baselines OccNet-1, OccNet-2 and OccNet-3 with MISE upsampling of 1, 2 and 3 times respectively. For fair comparison to other baselines, we use OccNet's refinement module to output its meshes with 5200 faces.

**Auto-encoding 3D shapes** We now evaluate NMF's ability to generate a shape given an input 3D point cloud and compare against AtlasNet [3] and AtlasNet-O[3] in Table 2. We note that NMF outperforms AtlasNet in terms of manifoldness with 20 times less self-intersections. NMF generates meshes with a higher normal consistency, leading to more realistic results in simulations and physically-based rendering. All the three methods have manifold vertices. While both NMF and AtlasNet-O have no non-manifold edges, AtlasNet yields a constant value of 7400 due to its constituent 25 non-manifold open templates. Visualizations in Fig. 6 show severe self-intersections and flipped normals for AtlasNet baselines which are absent for NMF. This leads to NMF giving more realistic physically based rendering results. Note the reflection of red box and green ball through NMF mesh, which are either distorted or absent for AtlasNet. The blue ball's reflection on conductor's surface is closer to ground truth for NMF due to higher manifoldness.

**Single-view reconstruction** We evaluate NMF for single-view reconstruction and compare against state-of-the-art methods in Table 3. We note significantly lower self-intersections for NMF compared

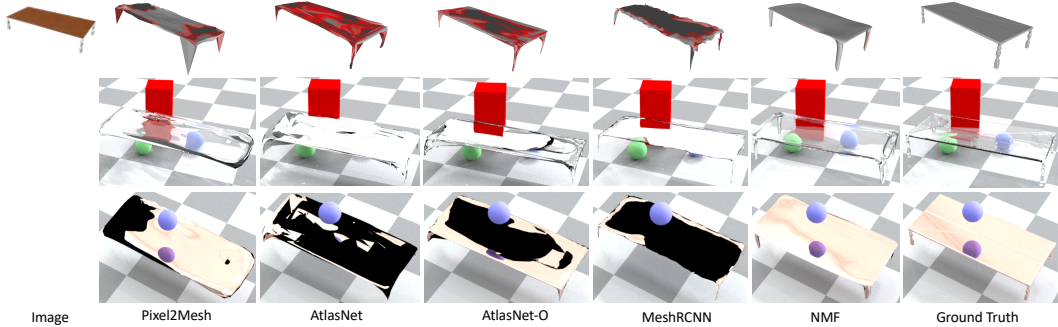

| | Image | Pixel2Mesh | AtlasNet | AtlasNet-O | MeshRCNN | NMF | Ground Truth |

Figure 7: Single View Reconstruction: We compare NMF to other mesh generating baselines for SVR. Top row shows mesh geometry along with self-intersections (red) and flipped normals (black). Physically based renders for dielectric and conductor material are shown in rows 2 and 3 respectively. Notice the reflection of checkerboard floor, occluded part of red box and balls are all visible through NMF render but not with other baselines. This is due to the presence of severe self-intersection and flipped normals. The reflection of blue ball on metallic table is more realistic for NMF than other methods.

| | Chamfer-L2 (↓) | w/ Laplace | Normal Consistency (↑) | w/ Laplace | NM-Vertices (↓) | w/ Laplace | NM-Edges (↓) | w/ Laplace | NM-Faces (↓) | w/ Laplace | Self-Intersection (↓) | w/ Laplace |
|---|---|---|---|---|---|---|---|---|---|---|---|---|
| MeshRCNN[2] | **4.73** | **5.96** | 0.698 | 0.758 | 9.32 | 9.32 | 17.88 | 17.88 | 5.18 | 0.86 | 7.07 | 1.41 |
| Pixel2Mesh[4] | 5.48 | 10.79 | 0.706 | 0.720 | 0.00 | 0.00 | 0.00 | 0.00 | 3.33 | 0.88 | 12.29 | 6.52 |
| AtlasNet-25[3] | 5.48 | 7.76 | 0.826 | 0.824 | 0.00 | 0.00 | 7400 | 7400 | 1.76 | 0.48 | 26.94 | 17.57 |
| AtlasNet-sph[3] | 6.67 | 7.35 | 0.838 | 0.836 | 0.00 | 0.00 | 0.00 | 0.00 | 2.19 | 1.08 | 11.07 | 5.94 |
| NMF | 7.82 | 8.64 | 0.829 | 0.837 | 0.00 | 0.00 | 0.00 | 0.00 | 0.83 | 0.45 | 0.12 | **0.00** |
| NMF-M | 9.05 | 8.73 | **0.839** | 0.838 | **0.00** | **0.00** | **0.00** | **0.00** | 0.76 | 0.42 | 0.11 | **0.00** |

Table 3: Single-view reconstruction.

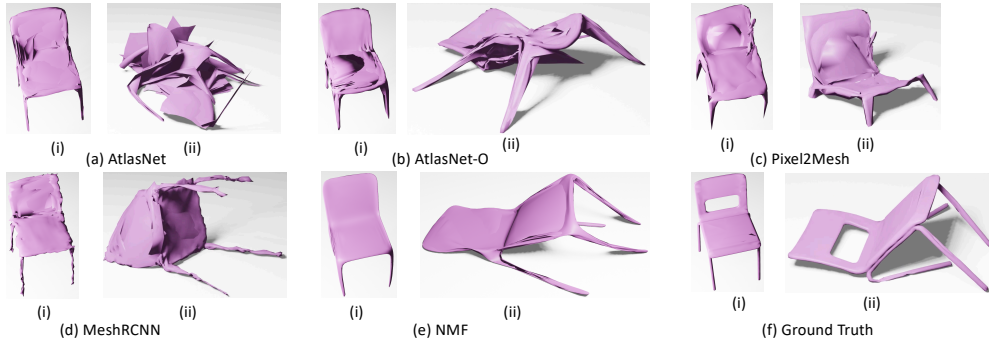

Figure 8: Qualitative results for soft body simulation. (a) While AtlasNet breaks down into 25 meshlets, (b) AtlasNet-O suffers from severe self-intersections leading to unrealistic simulations. (c) Pixel2Mesh leads to artifacts such as the chair going through the floor due to higher number of non-manifold faces. (d) MeshRCNN has a high degree of non-manifoldness resulting in unrealistic simulation. (e) NMF due to being a manifold mesh, is close to (f) the ground truth.

to the best baseline even after smoothing. Our method again results in fewer than 50% non-manifold faces compared to the best baseline. NMF-M also gets the highest normal consistency performance. Due to the cubify step as part of the MeshRCNN [2] pipeline which converts a voxel grid into a mesh, the method has several non-manifold vertices and edges compared to deformation based methods Pixel2Mesh [4, 2], AtlasNet-O [3] and NMF. AtlasNet suffers from the most number of non manifold edges, almost 100 times that of MeshRCNN. We note that MeshRCNN[2] better performance in Chamfer Distance come at a cost of other metrics. We qualitatively show the effects of non-manifoldness in Figure 7 and supplementary material. We observe that for dielectric material (second row), NMF is able to transmit background colors closest to the ground truth, whereas other baselines only reflect the white sky due to the presence of flipped normals.

**Soft body simulation (watch supplementary video for better understanding)** To further demonstrate the usefulness of manifoldness, we qualitatively evaluate predicted meshes with soft body simulation in Fig. 8. Here we simulate dropping meshes on the floor using Blender [57], with settings $pull = 0.9$, $push = 0.9$, $bending = 10$ to represent a rubber-like material. We note that AtlasNet [3] breaks into its constituent 25 independent meshlets upon hitting the floor. This behaviour is expected of methods that predict shapes as a set of n-connected components [58]. Both Pixel2Mesh

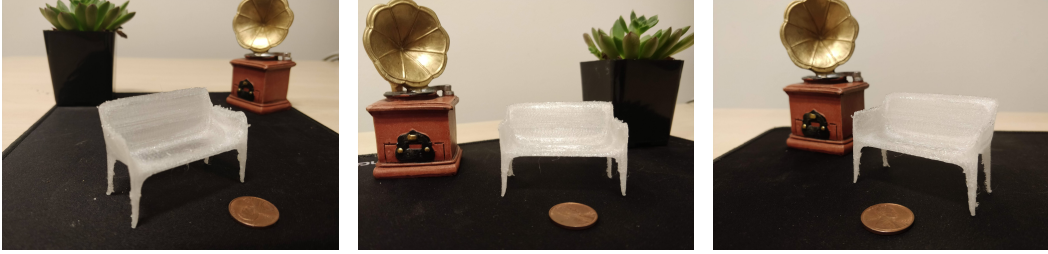

Figure 9: Few renderings of the 3D printed shape (bench) generated by NMF.

| Single View Recon. | Chamfer-L2 (↓) | Normal Consistency (↑) | NM-Vertices (↓) | NM-Edges (↓) | NM-Faces (↓) | Self-Intersection (↓) | Time (↓) |
|---|---|---|---|---|---|---|---|
| OccNet-1[8] | 8.77 | 0.814 | 1.13 | 0.85 | 0.36 | **0.00** | 871 |
| OccNet-2[8] | 8.66 | 0.814 | 2.67 | 1.79 | 0.21 | 0.03 | 1637 |
| OccNet-3[8] | 8.33 | 0.814 | 2.79 | 1.90 | **0.15** | 0.09 | 6652 |
| NMF | **7.82** | **0.829** | **0.00** | **0.00** | 0.83 | 0.12 | **187** |
| NMF w/ Laplace | 8.64 | 0.837 | 0.00 | 0.00 | 0.45 | **0.00** | 292 |

Table 4: Comparison with implicit representation method OccNet [8] for single view reconstruction.

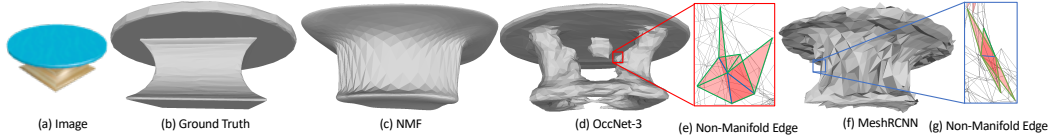

(a) Image    (b) Ground Truth    (c) NMF    (d) OccNet-3    (e) Non-Manifold Edge    (f) MeshRCNN    (g) Non-Manifold Edge

Figure 10: Implicit methods: OccNet fails to give meshes that are singly connected and MeshRCNN has poor normal consistency along with severe self-intersections. Both OccNet and NMF has non-manifold edges (shown as zoomed out insets). NMF generates meshes that are visually appealing with higher manifoldness.

[4] and AtlasNet-O [3] yield unrealistic simulations due to the presence of severe self-intersection artifacts. We note that MeshRCNN [2] suffers from *over-bounciness* due to non-manifoldness and poor normal consistency. In contrast, NMF yields simulations with properties that are closest to the ground truth.

**3D printing**   We now show in Fig. 9 a few renders of a 3D printed shape predicted by NMF using image from Figure 1. Since NMF predicts a manifold mesh, we can 3D print the predicted shapes without any post processing or repair efforts, obtaining satisfactory printed products.

**Comparison with implicit representation method**   We evalute NMF against state-of-the-art indirect mesh generation method OccNet [8] for the task of single view reconstruction in Table 4. We observe that NMF outperforms the best baseline OccNet-3 in terms of geometric accuracy. This is primarily because NMF predicts a singly connected mesh object as opposed to OccNet which leads to several disconnected meshes. Moreover, due to the limitations imposed by the marching cubes algorithm discussed in Section 2, OccNet-1,2,3 have several non-manifold vertices and edges where as by construction, NMF doesn't suffer from such limitation. An example of non-manifold edge is shown in Fig. 10. For sake of completeness, we also show the mesh generated by MeshRCNN [2] that suffers from non-manifold vertices and edges. NMF is also competitive with OccNet in terms of self-intersections since both methods become practically intersection-free with Laplacian smoothing . While OccNet outperforms NMF in terms of non-manifold faces, we argue that this comes at a cost of higher inference time. For reference, the fastest version of OccNet has comparable non-manifold faces and self-intersections but suffers relatively in terms of other metrics.

## 5   Conclusions

In this paper, we have considered the problem of generating manifold 3D meshes using point clouds or images as input. We define manifoldness properties that meshes must satisfy to be physically realizable and usable in practical applications such as rendering and simulations. We demonstrate that while prior works achieve high geometric accuracy, such manifoldness has previously not been sought or achieved. Our key insight is that manifoldness is conserved under a diffeomorphic flow that deforms a template mesh to the target shape, which can be modeled by exploiting properties of Neural ODEs [1]. We design a novel architecture, termed Neural Mesh Flow, composed of deformation blocks with instance normalization and refinement flows, to achieve manifold meshes without any post-processing. Our results in the paper and supplementary material demonstrate the significant benefits of NMF for real-world applications.

## Broader Impact

The broader positive impact of our work would be to inspire methods in computer graphics and associated industries such as gaming and animation, to generate meshes that require significantly less human intervention for rendering and simulation. The proposed NMF method addresses an important need that has not been adequately studied in a vast literature on 3D mesh generation. While NMF is a first step in addressing that need, it tends to produce meshes that are over-smooth (also reflected in other methods sometimes obtaining greater geometric accuracy), which might have potential negative impact in applications such as manufacturing. Our code, models and data will be publicly released to encourage further research in the community.

## Acknowledgement

We would like to thank Krishna Murthy Jatavallabhula and anonymous reviewers for valuable discussions and feedback. We would also like to thank Pengcheng Cao with UCSD CHEI for providing 3D printed models and Shreyam Natani for helping with Blender. This work was supported by NSF CAREER Award 1751365, along with generous gifts from Google and Adobe.

## Footnotes

[1] https://kunalmgupta.github.io/projects/NeuralMeshflow.html and try our colab notebook.

[2] In the scope of this work, meshes do not exhibit defects like duplicate elements, isolated vertices, degenerate faces and inner surfaces that can also cause a mesh to be *non-manifold*.

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
