[Supplementary Material]

# Supplementary Material for
# Neural Mesh Flow: 3D Manifold Mesh Generation via Diffeomorphic Flows

**Kunal Gupta    Manmohan Chandraker**
University of California, San Diego
{k5gupta, mkchandraker}@eng.ucsd.edu

## Abstract

In this **supplementary document**, we first give additional details about the experiments presented in **supplementary video** in Section 1. We then show additional qualitative results in Section 2. We present detailed results for experiments discussed in the main paper in section 3. We provide some additional details about our method in Section 4 and discuss its limitations and future scope in Section 5.

## 1   Supplementary Video

Physically based tasks like rendering, simulation and 3D printing require meshes to be manifold. Neural Mesh Flow learns to generate manifold meshes by construction since it models a diffeomorphic flow and thereby maintains the uniqueness and orientation preserving properties[1, 2]. However the other mesh generation methods AtlasNet[3], Pixel2Mesh[4, 5], MeshRCNN[5] and OccNet[6] fail to generate meshes that satisfy these *manifoldness* properties. In our supplementary video, we perform qualitative comparisons amongst the mesh generation methods for these physically based tasks.

### 1.1   Soft Body Simulation

One of the advantages of a manifold mesh is that it allows us to do physically based simulations. In this experiment, we specifically take the challenging task of simulating the dropping of a mesh on a floor. Amongst other things, this is a challenging task because all the mesh components must support the stress and strain on the mesh as a whole and should result in the solutions to dynamic equations that best represent the reality. The simulations were performed using [7] with settings $pull = 0.9, push = 0.9, bending = 10$ to represent a rubber like material.

We show the 3D meshes (i) and their final form after hitting the ground (ii) in figure 1. It is interesting to note that AtlasNet[3] (Fig 1 (a)) consisting of 25 mesh patches, while giving good geometric accuracy, disintegrates into independent parts since the collision dynamics are solved for each individual meshlet and therefore the results are far from the ground truth (Fig 1(f)). On the contrary AtlasNet-O[3] is able to retain the mesh structure but due to severe self-intersections, the collision simulation is unrealistic and the amount of self-intersections increase after hitting the ground, which shows that merely having the correct mesh geometry is not enough for physically realizable meshes, instead it should also have *manifoldness*. While Pixel2Mesh[4, 5](Fig 1 (c)) also suffers simulation artifacts from self-intersections, we note that its mesh contains very few and sparse set of vertices to represent important shape features (like legs). Because of this *non-manifoldness* we encounter strange simulation behaviours such as the legs going through the floor (Fig 1 (c)) which is unrealistic. MeshRCNN[**?** ]  is found to suffer from *over-bounciness* of its meshes during simulation. We believe this is because of its poor normal consistency which causes issues when solving contact force equations. Neural Mesh flow, (Fig 1(e)) due to its high manifoldness gives realistic simulations

Figure 1: Qualitative results for soft body simulation

that are close to the ground truth (Fig 1(f)) which demonstrates its effectiveness and reinforces our hypothesis that *manifoldness* is key to physically realizable meshes. We also tested simulation for OccNet [6] but found that it crashed the simulator due to the presence of several non-manifold vertices and faces.

## 1.2 Physically based renderings

Another application that greatly benefits from manifold meshes is physically based rendering. In this experiment we investigate how various meshes behave under reflection and refraction of light by rendering them with dielectric and conducting materials. While we compare against several baselines AtlasNet[3], AtlasNet-O[3], MeshRCNN[5] and Pixel2Mesh[4], we also improve their manifoldness by adopting a simple post-processing step involving three iterations of Laplacian smoothing. This is done to minimize the gap in manifoldness of NMF and other baselines. In Fig. 2 we show the mesh reconstruction as well as rendering results for all baselines with Laplacian smoothing, termed 'w/ Lap.' and without it. The self-intersecting triangles are shown in color red while flipped normals containing non-manifold faces is shown in color black. We notice that by applying Laplacian filter, the amount of self-intersections and flipped normals is reduced. However, this comes at a cost of geometric accuracy as can be seen for Pixel2Mesh where the legs of its chair are no longer present. This is in accordance to the *regularizer's dilemma*. Notice that NMF generates meshes with much higher manifoldness and therefore does not require any post-processing. For the case of dielectric rendering, we observe that AtlasNet baselines perform very poorly even with smoothing. While Pixel2Mesh lacks sufficient geometric accuracy post smoothing. MeshRCNN[5] does improve with smoothing but still has several non-manifold artifacts. Notice the image of red box and green ball which are distorted under MeshRCNN and Pixel2Mesh but is very accurate and clear with NMF and is infact closer to the ground truth. Even the checkerboard ground has quite realistic reflection with NMF as opposed to baselines. For conductor material, it is important to take note of reflection of the blue ball as well as self-reflections of the shape. We observe that due to the fliiped faces, all baselines fail to various degrees in getting correct reflection of ball. Moreover, we can observe

Figure 2: Physically based renderings

| 2D Image | NMF | Ground Truth |

Figure 3: Additional Physically based renderings from NMF

Figure 4: Example of a 3D printed shape generate by NMF

the self-reflection of the chair for NMF which is pretty close to the ground truth. We observe that Laplacian smoothing help remove isolated patches of non-manifoldness, but it fails to remove large patches without making significant losses in terms of geometric accuracy.

## 1.3 3D Printing

We show a few renders of a 3D printed shape. The shape was generated from NMF without performing any post processing to the prediction. It is important to note that printing other methods require significant human inputs owing to high degrees of non-manifold issues. Fig 4 shows a 3D printed bench that was generated by NMF. Not only does it aids rapid 3D printing technology, the results thus obtained are very satisfactory.

## 2 More Qualitative Results

In this section we show additional applications that are enabled by NMF without any changes to its architecture. These mainly include texture mapping, global shape parameterization, shape deformation and correspondence.

### 2.1 Texture Mapping and Parameterization

One of the important problems in graphics research is that of global shape parameterization which is often used to carry out texture mapping. Since NMF learns to diffeomorphically flow a spherical mesh to a target shape, it retains the local geodesics. This allows us to take a spherical texture (Fig 5(a)) and map it to generated shapes (Fig. 5(b-c)) without any human inputs. As we can qualitatively observe, the texture mapping is satisfactorily without any artifacts and distortions.

### 2.2 Shape Deformation, Interpolation

For any shape auto-encoder that strives to learn meaningful representations it is important to enable smooth latent space interpolations and have knowledge sharing across generated shapes. For the

(a) Texture          (b) Chair          (c) Car          (d) Airplane

Figure 5: Texture Mapping using NMF

Figure 6: Latent space interpolation of NMF

specific case of NMF that learns to diffeomorphically map a sphere to the target shape, given its shape embedding, the problems of shape deformation and latent space interpolation are identical. To this end, given two shape $M_0, M_1$ we feed them the point encoder to get their respective shape embeddings $z_0, z_1$. By linearly interpolating $z = \lambda z_0 + (1 - \lambda)z_1, \lambda \in [0, 1]$ we can get continuous and manifold intermediate shape deformations and interpolations. While we show several such deformations in out **supplementary video**, we illustrate a few more interpolations in Fig 6 where we observe cross category interpolations that retain manifoldness property at each intermediate step.

## 2.3  Semantic Correspondence

One of the consequence of having smooth interpolations is that NMF is able to learn part correspondence across intances in a category (Fig 7) as well as through instances belonging to different categories (Fig 8). It is important to mention that this is purely a consequence of NMF architecture and learning such semantic correspondences does not require any explicit training. In Fig. 7 (a) we note that the front and back part of the cars (including the wheels) have the same color which implies that they are semantically correlated. Similarly, the wings and tail of airplane (Fig7(c)) are semantically correlated among the two instances. Interestingly, for shapes where there is significant change in geometry (Fig 7(b,d)) such as a table/chair having four legs and not, we observe that NMF still maps the legs in the initial shape (top) to the base of the target shapes (bottom) that act as *pseudo-legs*. We observe NMF's ability to learn semantic correspondence even across categories (Fig 8). The legs of table (Fig 8(a)) are semantically mapped to the legs of a chair (Fig 8(b)) and even to a bench (Fig. 8(c)).

(a) Car        (b) Chair        (c) Airplane        (d) Table

Figure 7: Semantic correspondence learned by NMF without supervision

(a) Table        (b) Chair        (c) Bench

Figure 8: Cross-Category semantic correspondence learned by NMF without supervision

Thus, the above observations indicate that NMF learns really meaningful latent space for 3D manifold shapes.

# 3    Quantitative Results

We list down the exhaustive quantitative results for the task of auto-encoding (Table 1) and single view reconstruction (Table 2-7).

## 3.1    Comparison with Implicit Representation method

Here we show some more comparisons between OccNet[6] and NMF. We show quantitative results in Table 8 for the task of point cloud completion and single view reconstruction. We observe that while OccNet has several from non-manifold vertices and non-manifold edges, NMF is by construction is immune to such issues. NMF also achieves higher normal consistency which makes it better suited from physically based tasks like rendering and simulation. While both methods have can give meshes with almost zero self-intersections, we observe that OccNet does better in terms of non-manifold faces. However, we argue that this comes at a huge cost of inference time since the fastest variant of OccNet (almost 2.5 times slower than NMF) has similar score for NM-faces while performing significantly poor in other metrics. We also show rendering comparisons in Fig 9. Due to non-manifoldness, occnet does not provide sufficient rendering quality and NMF clearly is closer to the ground truth.

| | Chamfer-L2 (↓) | | | Normal Consistency (↑) | | | NM-Faces ↓ | | | Self-Intersection (↓) | | |
|---|---|---|---|---|---|---|---|---|---|---|---|---|
| | AtNet | AtNet-O | NMF | AtNet | AtNet-O | NMF | AtNet | AtNet-O | NMF | AtNet | AtNet-O | NMF |
| table | 8.65 | 4.38 | 7.08 | 0.837 | 0.842 | 0.853 | 2.10 | 1.96 | 1.18 | 28.22 | 9.42 | 0.20 |
| couch | 2.84 | 1.98 | 3.24 | 0.679 | 0.668 | 0.691 | 1.02 | 0.44 | 0.14 | 26.96 | 0.79 | 0.01 |
| speak. | 5.46 | 5.50 | 7.30 | 0.680 | 0.679 | 0.710 | 0.40 | 0.14 | 0.04 | 22.74 | 0.28 | 0.00 |
| firea. | 1.20 | 1.88 | 2.19 | 0.975 | 0.975 | 0.977 | 2.17 | 1.51 | 0.01 | 24.86 | 6.46 | 0.00 |
| plane | 1.11 | 1.20 | 2.76 | 0.938 | 0.939 | 0.957 | 2.96 | 2.51 | 0.91 | 22.75 | 10.06 | 0.04 |
| chair | 3.80 | 5.19 | 5.87 | 0.682 | 0.697 | 0.704 | 2.01 | 1.89 | 1.63 | 22.94 | 7.85 | 0.20 |
| monit. | 1.76 | 1.57 | 2.27 | 0.734 | 0.737 | 0.699 | 0.76 | 0.80 | 0.02 | 25.94 | 2.83 | 0.00 |
| phone | 1.74 | 1.40 | 2.36 | 0.910 | 0.765 | 0.765 | 0.23 | 0.05 | 0.02 | 23.98 | 0.16 | 0.00 |
| boat | 1.60 | 2.36 | 4.66 | 0.835 | 0.838 | 0.849 | 0.86 | 0.47 | 0.02 | 20.90 | 2.73 | 0.00 |
| lamp | 6.21 | 7.00 | 19.23 | 0.917 | 0.924 | 0.923 | 1.11 | 2.18 | 0.82 | 16.81 | 9.01 | 0.20 |
| bench | 2.13 | 1.81 | 3.02 | 0.917 | 0.917 | 0.926 | 2.41 | 1.65 | 0.60 | 24.17 | 6.95 | 0.06 |
| car | 3.00 | 2.74 | 3.51 | 0.770 | 0.781 | 0.805 | 1.37 | 0.49 | 0.08 | 30.67 | 1.43 | 0.01 |
| cabin. | 3.69 | 3.53 | 4.10 | 0.900 | 0.897 | 0.896 | 0.67 | 0.23 | 0.15 | 26.64 | 0.62 | 0.01 |
| mean | 4.15 | 3.50 | 5.54 | 0.815 | 0.816 | 0.826 | 1.72 | 1.43 | 0.71 | 24.80 | 6.03 | 0.10 |

Table 1: Auto Encoding Performance

| Category | table | couch | speak. | firea. | plane | chair | monit. | phone | boat | lamp | bench | car | cabin. | mean |
|---|---|---|---|---|---|---|---|---|---|---|---|---|---|---|
| MeshRCNN[5] | 5.34 | 3.73 | 8.27 | 2.07 | 2.27 | 5.56 | 4.17 | 3.00 | 3.55 | 13.67 | 3.21 | 3.33 | 5.11 | 4.73 |
| Pixel2Mesh[4] | 6.64 | 4.48 | 9.76 | 2.42 | 2.71 | 6.66 | 5.03 | 3.57 | 3.78 | 16.55 | 3.80 | 3.41 | 5.86 | 5.48 |
| AtlasNet-25[3] | 8.67 | 4.97 | 10.38 | 2.08 | 2.12 | 5.77 | 5.08 | 3.50 | 3.62 | 15.73 | 3.32 | 4.06 | 5.14 | 5.48 |
| AtlasNet-sph[3] | 8.59 | 6.57 | 12.27 | 3.06 | 2.48 | 8.167 | 8.29 | 4.47 | 4.97 | 17.63 | 4.50 | 4.29 | 4.65 | 6.67 |
| NMF (Ours) | 10.95 | 6.20 | 12.95 | 4.67 | 3.70 | 8.94 | 7.94 | 4.88 | 7.15 | 26.49 | 4.85 | 4.566 | 5.139 | 7.82 |

Table 2: Single View Reconstruction: Chamfer Distances

| Category | table | couch | speak. | firea. | plane | chair | monit. | phone | boat | lamp | bench | car | cabin. | mean |
|---|---|---|---|---|---|---|---|---|---|---|---|---|---|---|
| MeshRCNN[5] | 0.743 | 0.723 | 0.717 | 0.623 | 0.693 | 0.708 | 0.782 | 0.848 | 0.648 | 0.655 | 0.655 | 0.649 | 0.730 | 0.698 |
| Pixel2Mesh[4] | 0.723 | 0.743 | 0.761 | 0.612 | 0.685 | 0.703 | 0.805 | 0.843 | 0.680 | 0.643 | 0.654 | 0.683 | 0.745 | 0.706 |
| AtlasNet-25[3] | 0.813 | 0.787 | 0.786 | 0.969 | 0.958 | 0.725 | 0.680 | 0.755 | 0.871 | 0.918 | 0.898 | 0.835 | 0.778 | 0.826 |
| AtlasNet-sph[3] | 0.808 | 0.798 | 0.790 | 0.971 | 0.962 | 0.740 | 0.695 | 0.759 | 0.881 | 0.923 | 0.901 | 0.838 | 0.777 | 0.838 |
| NMF (Ours) | 0.844 | 0.783 | 0.792 | 0.971 | 0.963 | 0.739 | 0.696 | 0.755 | 0.881 | 0.925 | 0.932 | 0.829 | 0.778 | 0.829 |

Table 3: Single View Reconstruction: Normal Consistency

| Category | table | couch | speak. | firea. | plane | chair | monit. | phone | boat | lamp | bench | car | cabin. | mean |
|---|---|---|---|---|---|---|---|---|---|---|---|---|---|---|
| MeshRCNN[5] | 15.869 | 3.052 | 3.055 | 2.332 | 2.630 | 16.292 | 6.733 | 0.468 | 3.503 | 23.496 | 21.710 | 1.586 | 10.281 | 9.319 |
| Pixel2Mesh[4] | 0.0 | 0.0 | 0.0 | 0.0 | 0.0 | 0.0 | 0.0 | 0.0 | 0.0 | 0.0 | 0.0 | 0.0 | 0.0 | 0.0 |
| AtlasNet-25[3] | 0.0 | 0.0 | 0.0 | 0.0 | 0.0 | 0.0 | 0.0 | 0.0 | 0.0 | 0.0 | 0.0 | 0.0 | 0.0 | 0.0 |
| AtlasNet-sph[3] | 0.0 | 0.0 | 0.0 | 0.0 | 0.0 | 0.0 | 0.0 | 0.0 | 0.0 | 0.0 | 0.0 | 0.0 | 0.0 | 0.0 |
| NMF (Ours) | 0.0 | 0.0 | 0.0 | 0.0 | 0.0 | 0.0 | 0.0 | 0.0 | 0.0 | 0.0 | 0.0 | 0.0 | 0.0 | 0.0 |

Table 4: Single View Reconstruction: Manifold Vertices

| Category | table | couch | speak. | firea. | plane | chair | monit. | phone | boat | lamp | bench | car | cabin. | mean |
|---|---|---|---|---|---|---|---|---|---|---|---|---|---|---|
| MeshRCNN[5] | 34.317 | 7.644 | 7.202 | 9.294 | 24.357 | 25.037 | 10.161 | 2.599 | 8.652 | 34.398 | 31.411 | 4.030 | 12.924 | 17.783 |
| Pixel2Mesh[4] | 0.0 | 0.0 | 0.0 | 0.0 | 0.0 | 0.0 | 0.0 | 0.0 | 0.0 | 0.0 | 0.0 | 0.0 | 0.0 | 0.0 |
| AtlasNet-25[3] | 7.40e3 | 7.40e3 | 7.40e3 | 7.40e3 | 7.40e3 | 7.40e3 | 7.40e3 | 7.40e3 | 7.40e3 | 7.40e3 | 7.40e3 | 7.40e3 | 7.40e3 | 7.40e3 |
| AtlasNet-sph[3] | 0.0 | 0.0 | 0.0 | 0.0 | 0.0 | 0.0 | 0.0 | 0.0 | 0.0 | 0.0 | 0.0 | 0.0 | 0.0 | 0.0 |
| NMF (Ours) | 0.0 | 0.0 | 0.0 | 0.0 | 0.0 | 0.0 | 0.0 | 0.0 | 0.0 | 0.0 | 0.0 | 0.0 | 0.0 | 0.0 |

Table 5: Single View Reconstruction: Manifold Edges

| Category | table | couch | speak. | firea. | plane | chair | monit. | phone | boat | lamp | bench | car | cabin. | mean |
|---|---|---|---|---|---|---|---|---|---|---|---|---|---|---|
| MeshRCNN[5] | 5.29 | 3.69 | 3.27 | 7.47 | 7.74 | 5.38 | 3.21 | 1.19 | 5.77 | 7.15 | 5.74 | 2.80 | 3.10 | 5.18 |
| Pixel2Mesh[4] | 2.74 | 2.37 | 2.50 | 5.80 | 5.36 | 3.53 | 3.04 | 2.61 | 4.21 | 5.39 | 3.18 | 2.49 | 2.75 | 3.33 |
| AtlasNet-25[3] | 1.77 | 0.92 | 0.40 | 4.14 | 3.52 | 1.84 | 0.756 | 0.31 | 1.40 | 1.72 | 1.91 | 1.17 | 0.50 | 1.76 |
| AtlasNet-sph[3] | 2.45 | 1.17 | 0.88 | 4.23 | 3.01 | 2.54 | 1.80 | 0.73 | 2.73 | 3.50 | 2.27 | 0.67 | 0.51 | 2.19 |
| NMF (Ours) | 1.22 | 0.08 | 0.06 | 0.00 | 1.02 | 1.81 | 0.02 | 0.02 | 0.03 | 1.00 | 0.43 | 0.06 | 0.16 | 0.83 |

Table 6: Single View Reconstruction: Manifold Faces

| Category | table | couch | speak. | firea. | plane | chair | monit. | phone | boat | lamp | bench | car | cabin. | mean |
|---|---|---|---|---|---|---|---|---|---|---|---|---|---|---|
| Pixel2Mesh[4] | 10.18 | 10.41 | 9.81 | 17.27 | 18.26 | 12.07 | 12.24 | 10.87 | 15.16 | 16.57 | 11.21 | 11.34 | 10.06 | 12.29 |
| AtlasNet-25[3] | 29.94 | 28.11 | 25.46 | 32.10 | 29.82 | 24.69 | 26.69 | 27.23 | 26.30 | 19.48 | 27.83 | 30.14 | 27.60 | 26.94 |
| AtlasNet-sph[3] | 12.68 | 5.25 | 4.89 | 20.92 | 15.56 | 13.67 | 12.33 | 3.29 | 14.37 | 12.75 | 14.95 | 2.67 | 2.20 | 11.07 |
| NMF (Ours) | 0.24 | 0.00 | 0.00 | 0.00 | 0.07 | 0.26 | 0.00 | 0.00 | 0.00 | 0.26 | 0.05 | 0.00 | 0.01 | 0.12 |

Table 7: Single View Reconstruction: Self-Intersection

| Point Completion | Chamfer-L2 (↓) | Normal Consistency (↑) | NM-Vertices (↓) | NM-Edges (↓) | NM-Faces (↓) | Self-Intersection (↓) | Time (↓) |
|---|---|---|---|---|---|---|---|
| OccNet-1[8] | 8.77 | 0.804 | 1.13 | 0.85 | 0.36 | **0.00** | 795 |
| OccNet-2[8] | 2.82 | 0.804 | 5.00 | 3.75 | 0.28 | 0.02 | 1622 |
| OccNet-3[8] | **2.69** | 0.805 | 6.74 | 3.74 | **0.23** | 0.08 | 7973 |
| NMF | 5.53 | **0.826** | **0.00** | **0.00** | 0.71 | 0.10 | **189** |
| NMF w/ Laplace | 5.25 | 0.825 | 0.00 | 0.00 | 0.38 | **0.00** | 294 |
| Single View Recon. | Chamfer-L2 (↓) | Normal Consistency (↑) | NM-Vertices (↓) | NM-Edges (↓) | NM-Faces (↓) | Self-Intersection (↓) | Time (↓) |
| OccNet-1[8] | 8.77 | 0.814 | 1.13 | 0.85 | 0.36 | **0.00** | 871 |
| OccNet-2[8] | 8.66 | 0.814 | 2.67 | 1.79 | 0.21 | 0.03 | 1637 |
| OccNet-3[8] | 8.33 | 0.814 | 2.79 | 1.90 | **0.15** | 0.09 | 6652 |
| NMF | **7.82** | 0.829 | **0.00** | **0.00** | 0.83 | 0.12 | **187** |
| NMF w/ Laplace | 8.64 | 0.837 | 0.00 | 0.00 | 0.45 | **0.00** | 292 |

Table 8: Comparison with Implicit Representation method

(a) Image

(b) OccNet-3

(c) NMF

(d) Ground Truth

Figure 9: Renderings for OccNet

# 4 Ablation Study

In this section we provide some more details about the architecture of NMF and various design choices. On of the important hyperparameters for NeuralODE[9] is the *tolerance* value. This determines the step size of the ODE solver. In Table 9 we show the trend in geometric accuracy as well as manifoldness at various tolerance values. Clearly, by taking a lower value of tolerance, we get higher geometric accuracy as well as manifoldness, but this comes at a slight cost of inference time. In practice, we did not find much improvement in results by taking tolerance lesser than $1e^{-5}$.

In order to allow NMF to learn high resolution meshes across several categories, we introduce two novel modules in our architecture. These include the instance normalization layer and refinement modules. As can be observed form Table 10 without any instance normalization, the network struggles to learn accurate geometry and has very poor manifoldness (Fig 11). This can also be interpreted as the NODE being *strained* during the learning process. This is because unlike traditional MLPs, in a NODE, points on the sphere need to be *flown* to the target location along specific lines of integration. Therefore, non-uniformity in the path lengths due to variations in categories results in NODE learning very complex dynamics. Instance normalization layer separates the task of learning shape attributes from that of learning shape variances. As shown in Fig 10, when using instance normalization, NMF focuses on learning the geometry and other features (Fig 10(a)) whereas the instance normalization learns to scale it to match the target variance (Fig 10 (b)). As we can see this greatly simplifies the task of learning 3D shapes across several categories.

| Error Tolerance | Chamfer-L2 (↓) | Normal Consistency (↑) | NM-Faces (↓) | Self-Intersection (↓) | Time (↓) |
|---|---|---|---|---|---|
| 1e-3 | 8.09 | 0.832 | 0.859 | 0.43 | 187 |
| 1e-4 | 6.09 | 0.828 | 0.75 | 0.24 | 184 |
| 1e-5 | 5.535 | 0.826 | 0.71 | 0.10 | 189 |

Table 9: Ablation - ET

(a) NMF prediction before IN            (b) NMF prediction after IN

Figure 10: Instance Normalization separates the task of learning shape attributes and shape variances

No-Norm          Deform Block-1          Deform Block-2          Deform Block-3

Figure 11: Effect of IN and refinement modules

We also get better geometric accuracy when using refinement flows as show in Fig 11. It is important to note that we observe manifoldness metrics decrease by a small amount with more refinement modules. We believe this decrease in manifoldness is because in order to approximate sharp features, more complex dynamics need to be learned, which in turn demand a tighter tolerance value. In this work we did not choose an extremely tight tolerance so as to get a higher inference speed. For applications that require extremely high manifoldness, a tighter tolerance can be used.

| | Chamfer-L2 ($\downarrow$) | Normal Consistency ($\uparrow$) | NM-Faces ($\downarrow$) | Self-Intersection ($\downarrow$) | Time ($\downarrow$) |
|---|---|---|---|---|---|
| No Instance Norm | 7.04 | 0.831 | 2.82 | 2.96 | 183 |
| 1 Deform block | 5.53 | 0.826 | 0.37 | 0.03 | 68 |
| 2 Deform block | 5.70 | 0.826 | 0.38 | 0.03 | 124 |
| 3 Deform block | 5.535 | 0.826 | 0.71 | 0.10 | 189 |

Table 10: Ablation

## 5 Limitations and Future Scope

The major limitation of our method is its restriction to generating shapes of genus-0 which is undesirable as real world objects tend to be of high genus and possess complex topology. Using templates more representative of target shape may allow handling higher genus, perhaps at the cost of generalizability. Approaches that learn such templates as part of their pipeline like MeshRCNN [5] can be explored to further generalize NMF to more object categories. Combining NMF with contemporary works like SIREN did not yield good results in our preliminary experiments, but remains an area that requires further study. While we focus on manifold shapes here, future work should also incorporate texture or material generation along with shape reconstruction.