[Reviews · NeurIPS 2020]

Review 1

Summary and Contributions: This paper addresses the problem of 3D shape generation with a neural network. Recent explicit approaches (e.g., AtlasNet, Pixel2Mesh, etc.) have issues where the generated shape may self-intersect. The technical contribution is an architecture with neural ordinary differential equation (NODE) modules as the key component to deform a sphere template to a target (genus-0) shape. NODE has the benefit of implicitly enforcing points to not self-intersect. Moreover, the paper incorporates instance normalization on the 3D points within the proposed architecture, which shows practical benefits. The paper compares against explicit and implicit approaches for shape generation and demonstrates the benefits of the approach, particularly with respect to self-intersections and other “manifoldness” criteria.

Strengths: I like this paper - the formulation is novel and interesting and the results look compelling. Self-intersection is a significant issue for explicit approaches (e.g., AtlasNet, Pixel2Mesh), and this paper delivers on this hard task. The paper is well written, the evaluation with respect to competing approaches looks solid, and the related work is good. I also really like the simulations shown in the supplementary video!

Weaknesses: I have one quibble with this paper: 1. Point-shape feature and its impact on self-intersection - I’m not convinced that the approach guarantees that self-intersections do not occur due to the point-shape feature. From what I understand, a 3D point goes through a linear mapping (to 512D) and then is dot-multiplied by the shape encoding. However, a trivial linear map could be one where the 3D point is mapped to the first three dimensions (by a 3x3 identity) and then the remaining dimensions are mapped to zero. Then, the input looks exactly like the augmented NODE [9], which allows for self-intersections. Note that this analysis holds regardless of the dot-multiply. Moreover, I think this observation is reflected in the results in Table 1 where the self-intersection criteria is not exactly zero (without Laplacian regularization). If this analysis is correct, then I suggest to soften the claims in the paper (e.g., Fig 1(b), L59-60, L108-109) and include a discussion on this point. Note that I don’t think this point is a deal-breaker for this paper - I think the results are compelling enough on its own despite this limitation. Suggestions for improving the paper: 2. Given the observation above of the point-shape feature, it would be great to see an ablation where the point encoding is concatenated or added with the shape encoding (instead of dot-multiplied). Moreover, it would be great to see the 3D point passed through a sinusoidal encoding (similar to the transformers and nerf papers) to see if more fine details can appear. 3. I really like the simulations in the supplemental video! It would be great to include a discussion with respect to cvxnets, which also shows simulations for generated 3D shapes. 4. It would be great to have a small discussion on failure modes.

Correctness: I see no major issues with correctness outside of the issue pointed out above (not a deal-breaker).

Clarity: This paper was a pleasure to read. A few small suggestions: The tables in Figs 1(b) and 5(b) are too small. Fig 5(a) - it’s a bit awkward to refer to Fig 5(a)b, etc. I noticed small grammar/spelling typos throughout.

Relation to Prior Work: The related work is good.

Reproducibility: Yes

Additional Feedback: Final feedback: I still recommend to accept this paper after considering the other reviews and the rebuttal. I’m confused by the rebuttal’s response to my question about the point-shape feature. The ODE solver is not operating over the 3D point directly but over the linear-layer's 512D response with the 3D point as input. So I still think it’s an augmented NODE formulation - or am I missing something here? While this corner case may not appear in practice, I think it's important to document it clearly if it is indeed true. For the final paper version, I'd like to see: 1. An ablation where the point and shape features are concatenated (instead of dot-multiplied). 2. Clarification regarding whether the ODE solver is operating over the linear-layer's 512D response (and not the 3D point directly). If the solver is operating over the 512D response, then the text needs to clarify its claims and relationship to augmented NODE. 3. Inclusion of this reference dealing with self-intersections that appeared at CVPR 2020: Leveraging 2D Data to Learn Textured 3D Mesh Generation. Paul Henderson, Vagia Tsiminaki, Christoph H. Lampert. 4. Merging of the references [8] and [16].


Review 2

Summary and Contributions: The paper introduces a new 3D shape generation model that utilizes blocks of Neural ODE based flows to deform a spherical mesh template into the generated shape. The flow module provides implicit shape regularization that prevents self-intersection etc. The authors demonstrated that shapes generated by this approach exhibit better manifold properties than baseline approaches such as AtlasNet and Pix2Mesh.

Strengths: The idea of using learn flows to deform geometries with implicit regularization to prevent self intersections and other non-manifold outcomes is interesting. This first part of the idea is discussed in Niemeyer et al., 2019 (Occupancy flow) and the latter is discussed in a concurrent work: Jiang et al. 2020 (ShapeFlow: Learnable Deformations Among 3D Shapes). Such flows do provide nice implicit regularizations.

Weaknesses: (1) The novelty of this proposed approach is rather limited. To me it seems like a direct hybrid of Pix2Mesh + Occupancy Flow. (2) The proposed approach towards shape generation still suffers many fundamental limitations, including inability to generate shapes beyond genus-0. (3) It is unclear why the authors excluded a wide range of state-of-the-art 3D shape generation models from their comparisons. Implicit function based models are somewhat of a gold standard for shape generation, allowing flexible output topology with fine sharp details. The authors seem to have excluded these models simply because they do not directly output a manifold mesh, which is a poor argument, since by taking the zero level set of an implicit function via marching cubes, it is trivial to retrieve a manifold mesh that is guaranteed intersection free etc., with all the nice properties of a manifold mesh. (4) The authors produced many renderings of the geometries with translucent materials and balls and shapes that seem rather irrelevant to the proposed model.

Correctness: The described methodology is sound. Experimental evaluation lacks many important baselines that should be compared against (i.e., Occupancy Networks).

Clarity: The method is rather clearly described. However various issues: (1) The page margin seems to have been altered on page 2. The bottom text collides with the page index. (2) There are many redundant information presented, e.g., renderings of the generated shapes with glass materials etc. Texture generation does not seem to be any part of the proposed methodology. (3) The paper clearly lacks numerous important baselines (which I think the proposed method stand a slim chance of beating).

Relation to Prior Work: See the "weakness" section for further details.

Reproducibility: Yes

Additional Feedback:


Review 3

Summary and Contributions: In this paper authors address a task of 3D mesh generation. They describe several manifoldness properties that are necessary for the generated meshes to be usable for in applications such as rendering and physical simulations. The key insight of the work is that these properties could be efficiently preserved via diffeomorphic flow that gradually deforms template mesh into the target shape. In particular, Neural Mesh Flow model employs Neural ODE blocks with instance normalization to achieve preservation of manifoldness without explicit post-processing. The approach is validated experimentally on ShapeNet on shape generation and single-view reconstruction tasks, and outperforms several recent baselines.

Strengths: 1. Instead of using explicit mesh regularizers and MLPs for gradual mesh deformations authors make use of Neural ODE blocks, which by design preserve manifold properties and in a way perform implicit mesh regularization that discourage self-intersections while maintain topology. Note that this is unlike some of the previous work which apply explicit regularizations, which often decrease the reconstruction accuracy (whereas not applying regularizations at all can lead to meshes which have poor manifoldness properties). This work seems to be novel, as it seems to be the first approach that employs NODE in the context of generating meshes with good manifoldness properties. 2. Extensive experimental comparison shows that NMF outperform other popular approaches (MeshRCNN, Pixel2Mesh, Atlas-Net, OccNet) on shape reconstruction and single-view reconstruction tasks. 3. An ablation study on physics-based rendering shows that NMF generates more physically suitable meshes that behave similarly to the ground truth shapes (which confirms that the manifoldness is useful practice).

Weaknesses: 1. It seems that the authors actually do apply Laplacian normalization to the generated meshes, which makes the claim of "not applying any reguralization" somewhat weaker. 2. Although authors provide some intuition about instance normalization but it’s not fully clear how critical that choice is. Additional investigations / experiments on this topic could uncover new insights about this issue.

Correctness: The mathematical derivation of the method seems correct and the choice of baselines and dataset for the empirical evaluation is reasonable, and gives a good feeling of the benefits of the proposed approach.

Clarity: The paper is well-written and easy to follow. Authors provide concise and comprehensible introduction to NODE and explain their advantages in the context of 3D mesh generation.

Relation to Prior Work: The method presented in the paper is closely related both to Neural ODEs and 3D mesh generation methods (MeshRCNN, Pixel2Mesh, Atlas-Net, OccNet), and in fact joins those two classes of methods together. Some existing mesh generation methods employed similar intuitions for regularization of their outputs, however, they do it in an explicit ad-hoc form, which often leads to poor geometrical accuracy. NMF solves this issue using Neural ODEs which introduce similar regularizations implicitly, ultimately generating smoother meshes with better manifoldness. Authors do a reasonable job at describing the relation to existing methods.

Reproducibility: Yes

Additional Feedback: Typos / etc: - Small formatting issue on L67 (page #) - L146: Fig 5ad - (very minor) having the same format for all references will make formatting nicer) ---- Update after rebuttal / reviews: there are no major concerns raised by others, and I still think the paper should be accepted.


Review 4

Summary and Contributions: This paper introduced a novel algorithm for 3D mesh generation called Neural Mesh Flow (NMF). The algorithm learn to deform an input template mesh into a target while enforcing manifoldness. The paper also designs some new metrics to evaluate the proposed approach, and comparing with state-of-the-art approaches, NMF achieves quite favorable results.

Strengths: This is a very nice paper that gives new perspectives in 3D mesh reconstruction from images, as well as shape deformation. The formulation of the proposed NODE framework is novel, and makes a lot of sense in preserving manifoldness of the shape. The instance normalization also not only empirically makes sense but also is key to make the algorithm perform better. Experimental results contain very convincing visual outputs, as well as good quantitative scores for the proposed metrics.

Weaknesses: To make the paper stronger, I have some constructive suggestions (rather than "weaknesses") that i still hope authors to address: - The number of steps T this algorithm performs NODE seems to be pre-determined by the system. Is a larger or smaller T impact the performance in a negative way? - In table 2, it seems MeshRCNN (along with other approaches) is better in terms of Chamfer-L2 scores, but the visualization clearly shows the proposed approach is better. Is there any explanations, or is this metric not informative in general? - For such shape deformation approach, the initialization of the starting pose seems crucial in terms of generating final outputs. Will initializing the template shape to be something other than a sphere lead to better (or worse) performance? - It's beyond the scope of this paper, but a lot of real-world objects are not in genus-0 shapes. Can authors comment on how to generalize to those new objects?

Correctness: Yes.

Clarity: Yes.

Relation to Prior Work: Yes.

Reproducibility: Yes

Additional Feedback: I hope authors could keep the promise and release the code/data as stated in the abstract.

[Author Response · NeurIPS 2020]

# Paper ID: 7163
## Neural Mesh Flow: 3D Manifold Mesh Generation via Diffeomorphic Flows

We thank the reviewers for their thoughtful feedback. We are delighted that (**R1**, **R3**, **R4**) voted clear accept. We appreciate their finding the paper *"a pleasure to read"* (**R1**), with experiments comprising *"very convincing visual outputs, as well as good quantitative scores"* (**R4**). They find our formulation *"novel"* (**R3**), *"interesting"* (**R1**) and giving *"new perspectives in 3D mesh reconstruction"* (**R4**).

(**R2**) **Comparison to Occupancy Networks:** We believe **R2** missed comparisons to OccNet [16] included in the paper. NMF performs better on 6 out of 7 metrics as shown in Tab 3 and Fig 8, which we discuss in L265-279. Note that inference time of NMF is only 0.18 sec compared to 6.65 sec of OccNet – a speed up of 37x with superior results.

(**R2**) **"Zero level set of an implicit function via marching cubes** $\cdots$ **trivial to retrieve a manifold mesh"** We respectfully disagree. As discussed in L88-91, marching cubes involves "*rasterization of iso-surface* [which] *is a purely local operation*, [thus] *often leads to ambiguities*". In contrast, as stated in L271-273, NMF is designed to yield a manifold mesh. The fact that NMF outperforms OccNet in 3 out of 4 manifoldness metrics further validates our point.

(**R2**) **Relevance of simulations and physically based renderings:** Such illustrations are important to demonstrate that manifoldness is useful in practice, as also noted by **R1**, **R3** and **R4**. Both the visualizations in Fig 6-8 and the supplementary video convincingly demonstrate the advantages of NMF over SotA.

(**R2**) **Novelty:** This is the first work to explicitly define and guarantee manifoldness in mesh generation, allowing use in simulations without any post-processing while maintaining a good runtime performance. We note that all of **R1**, **R3**, **R4** rate the paper as novel and giving "*new perspectives in 3D mesh reconstruction*" (**R4**).

(**R2**) **ShapeFlow and OccFlow:** We thank **R2** for pointing us to concurrent work of ShapeFlow [Jiang et al. (2020)], which appeared on arXiv AFTER the NeurIPS deadline. Our contrast stated in L42-44 applies to all of ShapeFlow, OccFlow [15] and PointFlow [14] – they require category specific priors for reconstruction, while NMF does not. This makes NMF more generally applicable and trainable for a diverse set of shapes.

(**R1**) **Point-shape feature and its impact on self-intersection:** It is true that 3D point trajectories can intersect when the dynamics $f_\theta$ operates on a manifold in higher dimension containing augmented states [9]. However, NMF dynamics operate in 3D state-space and as such it does not cause self-intersections. Therefore, with any combination of point and shape features, the manifoldness will remain intact. Due to finite numerical precision, some self-intersections can occur, but vanish as $tolerance \rightarrow 0$ (see Table 9 in supplementary).

(**R3**, **R4**) **Ablations:** Instance normalization is crucial to the performance of NMF. We include its ablations in Fig. 5 and Fig 11 (supplementary). The integration time was empirically determined to be large enough for flow to work, but not too large to cause overfitting. We will add more discussion on these in the final version.

(**R3**) **Smoothing:** Please note that NMF meshes are rendered directly, without Laplacian smoothing. Other methods render poorly due to non-manifoldness (Fig 2 in supplementary), so only those methods are smoothed for visualizations.

(**R4**) **Chamfer-L2:** It is possible to get better Chamfer-L2 scores despite inaccurate topology, thus, we believe they should be complemented with manifoldness metrics to better judge mesh quality.

**Future work:** We thank **R1** for the intriguing idea of exploring [Sitzmann et al. (2020)] for further improvement. (**R2**) While we focus on manifold shapes here, our future work will incorporate texture generation along with reconstruction.

(**R4**) **Higher genus and templates** We agree that using templates more representative of target shape may allow handling higher genus, perhaps at the cost of generalizability. Approaches that learn such templates as part of their pipeline like MeshRCNN [2] can be explored to further generalize NMF to more object categories.

**Other feedbacks** We thank **R1** for highlighting common motivations of NMF and [Deng et al. (2020)] with respect to physics simulation. We will add a discussion on it as well as one on limitations such as higher genus shapes in the final version (**R4**). Any formatting issues and typos will also be addressed in the final version.

## References

Deng, B., Genova, K., Yazdani, S., Bouaziz, S., Hinton, G., and Tagliasacchi, A. (2020). Cvxnet: Learnable convex decomposition. In *Proceedings of the IEEE/CVF Conference on Computer Vision and Pattern Recognition*, pages 31–44.

Jiang, C., Huang, J., Tagliasacchi, A., Guibas, L., et al. (2020). ShapeFlow: Learnable deformations among 3D shapes. *arXiv preprint arXiv:2006.07982*.

Sitzmann, V., Martel, J. N. P., Bergman, A. W., Lindell, D. B., and Wetzstein, G. (2020). Implicit neural representations with periodic activation functions. *arXiv preprint arXiv:2006.09661*.


[Meta-Review · NeurIPS 2020]

3D shape generation with a neural net is studied. After a lengthy discussion all reviewers appreciated the contributions of the paper. This being said, some concerns remained which the reviewers wanted the authors to address (AC concurs): 1. An ablation where the point and shape features are concatenated (instead of dot-multiplied) in the final version. 2. Clarification regarding whether the ODE solver is operating over the linear-layer's 512D response (and not the 3D point directly). If the solver is operating over the 512D response, then the text needs to clarify its claims and relationship to augmented NODE. 3. Inclusion of this reference that appeared at CVPR: Leveraging 2D Data to Learn Textured 3D Mesh Generation. Paul Henderson, Vagia Tsiminaki, Christoph H. Lampert. 4. Merging of the references [8] and [16]. The camera ready should include those points.